# Characterization of Morphologically Distinct Components in the Tarsal Secretion of *Medauroidea extradentata* (Phasmatodea) Using Cryo-Scanning Electron Microscopy

**DOI:** 10.3390/biomimetics8050439

**Published:** 2023-09-20

**Authors:** Julian Thomas, Stanislav N. Gorb, Thies H. Büscher

**Affiliations:** Functional Morphology and Biomechanics, Institute of Zoology, Kiel University, Am Botanischen Garten 9, 24118 Kiel, Germany; sgorb@zoologie.uni-kiel.de (S.N.G.); tbuescher@zoologie.uni-kiel.de (T.H.B.)

**Keywords:** Phasmatodea, tarsal secretion, evaporation rate, adhesion, cryo-scanning electron microscopy

## Abstract

Attachment to the substrate is an important phenomenon that determines the survival of many organisms. Most insects utilize wet adhesion to support attachment, which is characterized by fluids that are secreted into the interface between the tarsus and the substrates. Previous research has investigated the composition and function of tarsal secretions of different insect groups, showing that the secretions are likely viscous emulsions that contribute to attachment by generating capillary and viscous adhesion, leveling surface roughness and providing self-cleaning of the adhesive systems. Details of the structural organization of these secretions are, however, largely unknown. Here, we analyzed footprints originating from the arolium and euplantulae of the stick insect *Medauroidea extradentata* using cryo-scanning electron microscopy (cryo-SEM) and white light interferometry (WLI). The secretion was investigated with cryo-SEM, revealing four morphologically distinguishable components. The 3D WLI measurements of the droplet shapes and volumes over time revealed distinctly different evaporation rates for different types of droplets. Our results indicate that the subfunctionalization of the tarsal secretion is facilitated by morphologically distinct components, which are likely a result of different proportions of components within the emulsion. Understanding these components and their functions may aid in gaining insights for developing adaptive and multifunctional biomimetic adhesive systems.

## 1. Introduction

Attachment to the substrate is an important phenomenon influencing the everyday lives of most insects, as it is used to accomplish different tasks, such as locomotion [1], resisting predators [2], supporting copulation [3], etc. Different attachment devices have evolved, which all follow various combinations of certain basic principles, to fulfil the function of attachment [4,5,6].

To generate attachment, most insect tarsi utilize adhesion supported by lateral shear (friction) [7]. Two types of tarsal attachment systems emerged in insects: hairy ones and smooth ones [6,8,9]. Both maximize the contact area with the substrate to increase the contribution of different physical forces (e.g., Van der Waals forces, capillary interactions, and viscous forces) and, consequently, adhesion [7,10,11,12]. In a smooth attachment system, the contact surface maximization is caused by the soft material of adhesive pads. It is hierarchically organized internally towards the pad surface with progressively splitting fibers or integral foams, resulting in a rather smooth and flexible surface, and enabling the replication of the substrate profile and the maximization of the actual contact area [7,13]. To support the performance of the adhesive system, most insects utilize wet adhesion, meaning they secrete a tarsal fluid into the contact interface [7].

Besides insects, other animal groups also produce a fluid onto the substrate to support the adhesion process [14,15,16]. However, the mechanisms can largely differ from those employed by insects. Tree frogs similarly, but convergently, produce an adhesive liquid in their mucus glands to strengthen their attachment [17,18]. Geckos do not produce large amounts of tarsal secretion, but it has been shown that a nanometer thin lipid film covers the surface of their adhesive setae [19]. Especially in marine environments, other depletion mechanisms are found: Echinoderms, for example, secrete a two-phase secretion, where the first phase generates the adhesion and the second phase dissolves the first phase [20,21,22,23]. Freshwater polyps (*Hydra* spp.) can achieve temporary adhesion based on their adhesive secretion, which consists mainly of fibers [24]. The fluid depletion of stick and leaf insects, in contrast, works differently: it is most likely a passive delivery mechanism of tarsal fluid from the adhesive pads onto the substrate that is facilitated by pressure and intermolecular forces [12].

A histological investigation of the smooth tarsal adhesive organs of *Gromphadorhina portentosa* (Blattodea) revealed that the tarsal secretion is produced by exocrine cells into a storage volume underneath the outer part of the endocuticle. It is then released through channels onto the substrate [25,26]. The chemical compositions and functions of the tarsal fluids of insects have been investigated in several studies during the past decades (e.g., [2,27,28,29,30,31,32,33,34,35]). A chemical analysis of the tarsal secretion was conducted for representatives of different insect groups, such as Diptera [36], Hymenoptera [37,38], Coleoptera [39,40,41,42,43,44,45,46], Orthoptera [28,47], and Blattodea [34]. These analyses show that the chemical composition of the secretion differs between insect groups. The components found in the tarsal secretion include a water-soluble part and a lipid-soluble part. The water-soluble substances include alcohols, glucose and other saccharides, amino acids, unipolar carbohydrates, polar proteins, and peptides. The lipid-soluble substances include hydrocarbons, fatty acids (saturated and unsaturated with a chain length between C16 and C20 in both free and glyceride forms), and true waxes [28,34,42,47]. Based on the chemical analysis conducted by Vötsch et al., (2002), it was concluded that the tarsal secretion of *Locusta migratoria* (Orthoptera) is a highly viscous emulsion consisting of lipid droplets in a water-like solution [28]. The identified differences in the composition suggest that the functions of the tarsal fluids and their mixtures can also differ in detail.

Experiments investigating the functions of tarsal secretions in several insect groups demonstrated that the fluid could support three main functions as follows:(I)It can increase the adhesion to a broad range of substrates. Experiments on the attachment performance of Phasmatodea [48] and Coleoptera [49], where the volume of the tarsal secretion was diminished through consecutive steps or by porous substrates, showed that the attachment forces were enhanced on smooth surfaces, but were reduced on rough surfaces, indicating that the fluid is a crucial part of attachment generation on rough surfaces [48,49]. This effect was additionally supported by experiments on the bioinspired micropatterned samples [50]. These results indicate that the secretion can fill the asperities of non-smooth substrates, thus increasing the real contact area and thereby the attachment forces [32,48,51,52,53]. The immersion of nanometric beads in the accumulated tarsal secretions of the beetle *Coccinella septempunctata* and the fly *Calliphora vicina* indicated different viscosities of 21.8 and 10.9 mPa × s, respectively, showing that the physical properties of the fluid diverge between the species [33]. The presence of the liquid in contact is expected to provide capillary forces that increase adhesion [4,7]. Additionally, the high viscosity of the fluid likely implements viscous forces and thereby increases attachment [54].(II)It contributes to decontamination. Contaminating the adhesive pads of the stick insect *Carausius morosus* with polystyrene beads and manipulating the amount of adhesive fluid showed that a high amount of fluid led to a faster recovery rate of adhesion than a low fluid amount. Thus, it is an important part of the self-cleaning mechanism of smooth adhesive pads [30,31].(III)It can compensate for different surface chemistry of substrates. Chemical analyses of the fluids allow for the interpretation of the interaction with different surfaces. Due to the presence of two phases (water-soluble and lipid-soluble phases), the emulsion should improve the attachment to hydrophilic and hydrophobic surfaces, as it acts as a coupling agent between the pad and substrates with different free surface energies [7,28,52,55].

Some of the components of the tarsal secretion resemble those found on the surface of the insect cuticle, potentially helping to reduce the evaporation rate of water through the adhesive pad and assisting in communication [38,56]. An investigation of the ultrastructure and frictional properties of the smooth pad of *Tettigonia viridissima* (Orthoptera) revealed that the fluid within the pad contributes to its viscoelastic behavior and the frictional forces subjected to the substrate [57,58]. As highlighted by these experiments and chemical analyses, it is evident that the tarsal secretion supports and affects locomotion and attachment, and therefore insect behavior. It is also apparent that these results show that the physical and chemical properties of the tarsal fluids differ greatly between species.

Despite considerable insights into the compositions and functions of tarsal secretions, approaches to investigate the details of fluid depletion in insects are scarce. Particularly, details of the interactions of the tarsal fluids with the substrates and between different components of the fluid on the surface of the attachment pads and on the substrate remain unexplored. Insects with large smooth attachment pads possess large areas that need to be covered by these secretions and are particularly prone to contamination.

We therefore analyzed the morphological characteristics of the footprint residues of the stick insect *Medauroidea extradentata*. Stick insects are among the largest insects [59] and they possess smooth adhesive pads [13,60,61,62], which are rather voluminous [55,63], and therefore should produce a significant volume of tarsal secretion. Stick insects have two types of smooth adhesive pads: the arolium and the euplantulae [13,64,65]. The arolium is situated on the pretarsus between the two claws and is mainly used to generate an adhesion force (force perpendicular to the surface), whereas the euplantulae are situated on the tarsomeres and contribute to friction (force horizontal to the surface) (Figure 1) [6,62,65,66,67]. We used cryo-scanning electron microscopy to analyze frozen footprints at a high magnification in their quasi-native (frozen) states. In addition, white light interferometry was used to measure the change in the volume of individual liquid components over time and quantify their evaporation rates. Through this combination of approaches, we aimed to investigate the structural and physical properties of the footprints left by both types of smooth attachment pads of this species. The findings may provide useful information (1) to understand adhesion in stick insects and (2) to enhance advances in the field of biomimetic multifunctional adhesives.

## 2. Materials and Methods

### 2.1. Animals

We used the phasmid species *Medauroidea extradentata* (Brunner von Wattenwyl, 1907) (Figure 1A) because of the presence of a broad range of data on the functional morphology and biomechanics of its tarsal attachment system [13,55,62,68].

The morphology of arolium and euplantulae represents the most common and least derived setup among phasmids with smooth adhesive microstructures on both attachment pads, without micro-ornamentation [63] (Figure 1B). Individuals were obtained from the laboratory cultures of the Department of Functional Morphology and Biomechanics (Kiel University, Kiel, Germany). The insects were fed with blackberry leaves ad libitium and kept in a regular day and night cycle. Only adult individuals with clean and intact legs were selected. The insects were kept with blackberry leaves in clean hard plastic boxes to reduce contamination of the adhesive pads.

### 2.2. Footprint Collection

Microscope slides (76 × 26 mm) and glass coverslips (12 mm) (Thermo scientific, Budapest, Hungary) were used as sampling substrates for investigation using white light interferometry (WLI) and cryo-scanning electron microscopy (cryo-SEM). The glass surfaces were thoroughly cleaned with the following protocol prior to sampling footprints (Figure 1C1): (1) 15 min in an ultrasonic bath with distilled water and soap (neutral intensive cleaner); (2) 15 min in an ultrasonic bath with distilled water; (3) 15 min in an ultrasonic bath with 100% pure ethanol; and (4) 1 h in vacuum in a desiccator.

To obtain a footprint, the insects were first anaesthetized with CO_2_ for 20 s. The tarsus was placed on a carefully cleaned glass slide or glass coverslip within a marked area. Glass slides for cryo-SEM investigation were previously sputter-coated with a 20 nm layer of gold–palladium (Figure 1C2). A second cleaned glass slide was placed on the dorsal side of the tarsus and pressed for 5 s with even pressure, and the leg was simultaneously pulled to generate some shear forces (Figure 1C3). Lastly, the second glass slide and the foot were carefully removed, and alteration of the footprint was avoided (Figure 1C4). The footprints were immediately used for investigation in WLI and cryo-SEM (Figure 1C5). The glass slides were stored in a closed glass chamber at 20.6–22.9 °C room temperature and 43.2–51.4% ambient humidity (measured within the chamber).

### 2.3. Cryo-Scanning Electron Microscopy

Fresh footprints were sampled on cleaned glass coverslips that were previously sputter-coated with 20 nm gold–palladium (Figure 1C).

The glass coverslips with the fluid footprints were mounted on aluminum stubs and carefully immersed in liquid nitrogen for 5 s. The footprint was then transferred into the cryo-preparation chamber (Gatan ALTO-2500 cryo-preparation system, Gatan, Abingdon, UK) at −140 °C of the SEM Hitachi S-4800 (HitachiHigh-Technologies, Tokyo, Japan). The frozen footprints were then observed in the SEM at −120 °C at an accelerating voltage of 3 kV without sputter coating. Subsequently, each sample was sputter-coated with gold–palladium (layer thickness 10 nm) in the preparation chamber at −140 °C and observed again at −120 °C with 3 kV accelerating voltage. Sputter coating was used to enhance the visualization of the surface structure of the footprint components. For some frozen samples, sublimation (freezing-drying) at −80 °C in the prechamber was performed prior to observations. Contrast adjustment and image cropping were performed using the software Photoshop CS6 (Adobe Systems Inc., San Jose, CA, USA).

### 2.4. White Light Interferometry (WLI)

The glass slides with the footprints were examined in the white light interferometer New View 6000 (Zygo, Darmstadt, Germany) and analyzed using the software MetroPro (Zygo, Middlefield, CT, USA). The glass slides were placed under the WLI and measured at 20.6–22.9 °C room temperature and 61–64.3% ambient humidity. Areas of interest, where enough of the fluid accumulated to form a measurable droplet, were selected with the build-in-mask function (Figure 1C5). The droplet volume and its change over time (rate of evaporation) were measured. The droplet volume was measured at the beginning and afterwards and was remeasured every day for at least 14 days. If the drops still showed measurable volume after 14 days, the measurements were continued. Each measurement series consisted of three measurements, which were performed at an interval of about three minutes; for the analysis, the mean value of the three measurements was subsequently used. In the case of strong changes in volume, the respective mask was adjusted accordingly. In some rare cases, it was observed that water vapor likely accumulated in the footprints and thereby increased their volumes. Footprints with an accumulation of water were excluded from the analysis.

To determine the evaporation rate of the droplets, the initial volume (day 0) was considered to be 100%, and the change in volume percentage over minutes (%/min) and days (%/day) was determined (for raw data see Appendix A). The data were statistically analyzed using R (R version 4.2.3, R Core Team, Vienna, Austria). For statistical analysis, the evaporation rates were compared with a Kruskal–Wallis one-way analysis of variance (ANOVA) on ranks, followed by Dunn’s post hoc test, since they were not normally distributed (Shapiro–Wilk test) and showed no homoscedasticity (Levene’s test).

### 2.5. Temperature and Ambient Humidity Measurements

Temperature and the ambient humidity were measured with a Tinytag Plus 2 TGP—4500 (Gemini Data Loggers, Chichester, UK) and analyzed using the software TinyTag Explorer 6.0 (Gemini Data Loggers, Chichester, UK). For the measurements in the closed glass chamber, 250 measurements at intervals of 30 min were conducted, and for the measurements at the WLI 200, measurements at intervals of 1 min were conducted.

## 3. Results

### 3.1. Analysis of Frozen Footprints

We analyzed the appearance of the fluid and solid residuals resulting from the contact of the tarsus with the glass surface. Cryo-SEM enabled us to visualize the components in their frozen state at −120 °C with a high magnification. Immediate freezing with liquid nitrogen after deposition allowed us to investigate the footprints in the condition just after fluid depletion. The micrographs, hence, show a temporary impression of the footprint at the time of its application (Figure 2).

The grey scale on the cryo-SEM images is influenced by two factors: the distance of the footprint to the detector and the electron density of the secretion. Both factors are affected by the thickness and conductivity of the secretion itself. The electron density is additionally influenced by the fluid’s composition. Accordingly, thin liquid layers with a low electron density are displayed brighter than the background, as well as the thick layers with a high electron density that appear dark (Figure 2). After sputter coating, the differences in electron density (conductivity) vanished due to the coverage by the gold–palladium sputtering (see Appendix A). An observation of the frozen footprints revealed distinct components, which differed in their morphology and the site of occurrence. These can generally be divided into four groups that can be distinguished based on their shape, size, surface structure, and site of occurrence. In Figure 2A, an overview of the imprint of an arolium of *M. extradentata* is shown with representations of all four components (Figure 2B–E). The footprint components include droplets (Figure 2B), flakes (Figure 2C), thin films (Figure 2D), and thick films (Figure 2E). These components, their exact distribution within a footprint, as well as the characteristics shared between the groups, will be described in detail below.

### 3.2. Distribution of the Tarsal Secretion and Solid Bodies within the Footprints

Footprints are characterized as all structures that are found in the vicinity of the application site of the adhesive pads and are morphologically different from the sputtered glass surface. They include the putative components of the secretion that support attachment, known as solid bodies, which presumably originate from the solidified tarsal secretion and environmental contaminations. The attachment pads of the tarsus and pretarsus leave distinct imprints in terms of their position and shape (Figure 3) in most cases, which enable differentiation of the origin of the residuals. The arolium usually leaves large, single-surface impressions, which comply with the shape of the adhesive pad with diameters between 500 µm and 1 mm (Figure 3A,B). The imprints of the euplantulae generally possess diameters between 100 and 200 µm and are situated 100–200 µm apart from one another (Figure 3C,D). Additional imprints were often observed between the two pad imprints that had an elongated form with lengths of around 400 µm (Figure 3D).

A footprint generally consists of a mixture of all four morphologically distinct components as well as contaminants (Figure 2). Usually, the thin film is the most common component, followed by thick films and droplets, while the flakes are the least frequently observed. Contaminants are usually covered by a film, which, in the case of small contaminants, causes them to aggregate with others to form large clusters (see below). The largest amount of residues were found at the edge of the adhesive pads’ imprints, which was especially noticeable in the footprint areas left by the arolium (Figure 3A,B). It was observed that some imprints can consist of only one or two components (Figure 3D imprints mainly consist of thin and thick films).

### 3.3. Droplets

Droplet components have a round, compact, and voluminous shape (Figure 4). Droplets were observed as single components (2–10 µm) (Figure 4C,F), as well as in larger complexes (30–100 µm in diameter) (Figure 4A,B). Different surface structures were observed on droplets. These can be either smooth, rough, or covered with nano-droplets (Figure 4C,E (smooth), F (rough), and D (nano-droplets)). When sputtered, no layered surface is visible in contrast to the other footprint components (see Appendix A). These components were found throughout the whole footprint and were unrelated to the position of the attachment pad.

### 3.4. Flakes

Flakes are components that can be observed individually (Figure 5A) and have a length between a few µm (Figure 5F) and 50 µm (Figure 5B). They have compressed (Figure 5C) or elongated shapes (Figure 5D). After sputtering, the flakes show a structured surface consisting of several parallel thin layers with a thickness of a few nanometers, which do not have a uniform shape (Figure 5E,F). Flakes were more frequently found in the arolium imprints than in the euplantulae imprints. In the imprints of both types of attachment structures, they were deposited at the edges of the majority of observed footprints (Figure 5A). Flakes were less often observed compared to the other tarsal footprint components.

### 3.5. Thin Films

The components that are displayed brightly in the cryo-SEM images, either because of their low volume or low electron density, and that cover large surface areas are classified as thin films (Figure 6). These components were observed in each imprint of the arolium and euplantulae (Figure 6B,C). The structures of the thin films differed depending on whether they were situated close to the other components or not. When the films were near the other components, we observed that the thin films formed a structure that covers a large area (Figure 6A–C). With an increasing distance to the other components, the thin films increasingly aggregated into smaller circles down to a few nanometers (Figure 6C,D,F). The thin films covered the largest areas in the observed footprints, with areas between 100 µm and 1 mm in diameter (Figure 6A,B). The surface structure was not discernible without sputtering (Figure 6E,F). Sputter coating of the thin films covered them completely and made them invisible, as the thickness of the coating was likely higher than the films themselves.

### 3.6. Thick Films

Thick films are components that are dark in the cryo-SEM images and thus have either a large volume and/or a high electron density (Figure 7A). They were found in the majority of footprint samples studied. In addition, these components exhibit a smooth surface structure before sputtering (Figure 7C) and form a coherent coverage of up to 100 µm in diameter, which resembles a thick liquid film (Figure 7B). These films can be uniform or show some gaps (Figure 7D,E). Individual isolated portions, possessing the above-mentioned appearance, were also observed and are classified as part of the thick film. These single units possessed no gaps and covered areas of only a few µm (Figure 7F). When sputtered, some thick films showed rough or granular surface structures (Figure 7B and Figure 8C,F).

In some samples, iced water was observed close to the thick film components. The crystalline water could envelop the thick film (Figure 8A), form individual circles on the surface (Figure 8B), or cover the surface as a network (Figure 8D). Thick films were observed in both the arolium and euplantula imprints. While they seemed to be evenly distributed in the euplantulae (Figure 7A), they were found more frequent at the edges in the arolium impressions (Figure 3B).

### 3.7. Contaminations

A wide variety of contaminations were found in the phasmid footprints, ranging from large contaminations (around 100 µm in width) (Figure 9A) to small 3 µm wide particles (Figure 9B). Usually, they were covered by the tarsal secretion (Figure 9C,D). The contaminations were mostly observed at the edge of the impressions. We detected that large contaminations were often isolated from the others (Figure 9A), whereas small contaminations were combined into large clusters by the tarsal secretion (Figure 9B–D). In addition to the contamination being coated by the adhesive secretion, it was also observed that they adhered to the surface of residuals (Figure 9E,F).

### 3.8. Evaporation Rates

A measurement of the change in volume over time (evaporation rate) of 68 fluid droplets from 25 footprints was performed over a period of up to 75 days (Figure 10). The change in the droplet volume over the period of measured days and the evaporation rate (in %/min) of the three droplet types are visualized in Figure 11A,B.

The droplets revealed evaporation rates with different slopes. Based on their evaporation behavior and rates, we distinguished three different types of droplets indicated by different colors in Figure 11A. These three types are (1) the non-evaporating droplets (yellow), whose volume did not change over the entire measured period; (2) the slowly evaporating droplets (blue), which showed a slow evaporation rate over the measured period; and (3) the fast-evaporating droplets (red), which displayed a fast evaporation rate and completely evaporated after a few days, or at maximum, 32 days.

The non-evaporating droplets showed an evaporation rate, which is the change in the droplet volume over the measured days, that was lower than 0.34%/day. In total, 14 droplets were assigned to this type, which showed mean evaporation rates of 0.0531 ± 0.283%/day and 0.0000369 ± 0.000196%/min (Figure 10A and Figure 11).

The slowly evaporating type included all droplets with an evaporation rate between 0.34%/day and 3.2%/day. The group contained 12 droplets and displayed mean evaporation rates of 1.075 ± 0.479%/day and 0.000747 ± 0.000333%/min (Figure 10B and Figure 11).

The fast-evaporating type showed a faster volume loss and included all droplets with an evaporation rate higher than 3.2%/day. This type was present in 44 droplets and exhibited mean evaporation rates of 8.629 ± 7.503%/day and 0.00599 ± 0.00521%/min (Figure 10C and Figure 11).

A statistical comparison between the evaporation rates of the three types showed that the evaporation rate of the fast-evaporating type was significantly higher than that of the other types (Dunn’s post hoc test, *p* < 0.001), whereas there was no significant difference between the evaporation rates of the slowly evaporating and non-evaporating types (Dunn’s post hoc test, *p* = 0.326) (Kruskal–Wallis one-way ANOVA on ranks, H = 50.912, d.f. = 2, *N* (fast-evaporating) = 44, *N* (slowly evaporating) = 12, *N* (non-evaporating) = 14).

### 3.9. Light Microscopy Observations

Randomly selected glass slides with deposited footprints were observed and filmed at different time intervals under an inverted microscope. During the observation, a fine needle was carefully pulled through individual droplets and the different behaviors of the droplets were observed and filmed (Appendix A). Two different droplet responses were observed: (1) droplets that appeared to be liquid and were split into smaller droplets by the needle (Appendix A) and (2) droplets that appeared to be more viscous, whereby the needle scratched their surface (Appendix A).

## 4. Discussion

The investigations of the tarsal secretion of *Medauroidea extradentata* using cryo-SEM and WLI revealed that the fluid contains morphologically more diverse components than previously assumed. Previous research on the morphological and physical properties of the tarsal secretion of species with large smooth attachment pads is rare due to its higher viscosity when compared to the tarsal secretion of flies and beetles [33,69]. Peisker et al., (2014) were able to measure the viscosity of the tarsal secretion of flies and beetles by using the Brownian motion of micro-beads within the fluid [33]. Their lower viscosity enabled them to accumulate enough fluid, but this method could not be applied to the tarsal secretion of phasmids. Firstly, due to the high viscosity, it was not possible to collect a large amount of tarsal fluid using self-pulled glass needles and a micromanipulator, and secondly, the tarsal fluid was distributed over the surface in numerous fine droplets, and thus, there was no sufficient initial volume (own observations). Nevertheless, some results regarding the physical properties of the tarsal secretion of Phasmatodea were recently provided in [35]. They measured the contact angle (°) and dewetting speed (µm/s) of the tarsal secretion of phasmids with different body sizes and showed that the surface tension and viscosity of the fluid on glass are independent of the body size [35]. The high variance of the contact angle and dewetting speed suggests that the tarsal secretion is not perfectly homogenous and likely consists of multiple physically distinct components, which is confirmed by our results.

We discovered that the fluid consists of at least four morphologically different parts (droplets, flakes, thin films, and thick films (Figure 2)) including liquid components with different physical properties (non-evaporating, slowly evaporating, and fast-evaporating droplets) (Figure 10 and Figure 11). These various constituents could explain the diverse properties of the tarsal secretions in the smooth attachment devices of Ensifera [58] that likely have similar secretions to that of the representatives of Phasmatodea. However, differences in the composition and quantity of the components may account for the differences in the fluid properties of different species. The complexity of biological tarsal fluid makes it difficult to reproduce the properties of this secretion using artificial-hydrocarbon-based components because the artificial fluid only mimics a part of this complex mixture, which might be surface-specific. Additionally, the mimicking quality of the other physical properties of natural secretions in biomimetic adhesive fluids remains unknown [70].

Sometimes, an additional imprint with a similar composition as those from the pads appeared between the imprints of the two euplantulae (Figure 3A,D). This is an interesting observation, which deserves mentioning here. Since there are fields of setae situated between them (Figure 1B), there are two possible origins of this imprint. First, the fluid potentially originates from the adhesive pad and is transported onto the hairs. Second, the hairs themselves secrete the fluid and therefore contribute to attachment.

### 4.1. Possible Origin of the Flake Component

The cryo-SEM allowed us to observe the tarsal fluid at a high resolution and thus identify four morphologically distinct structures (Figure 2). The stage temperature of −120 °C affected the behavior of the secretion, causing all of the previously liquid components to solidify. The solidification allowed us to identify the structural characteristics of each component. In order to obtain information about the actual behavior of the tarsal fluid, we included light microscopy observations and measurements of evaporation rates using the WLI (Figure 10 and Figure 11). The light microscopy observations revealed that the tarsal secretion consists of different parts with varying viscosity, with few droplets hardening after time passes. The varying viscosities were detected by drawing a fine needle through individual droplets (see light microscopy observations). In some droplets, a scratching of the surface was observed, indicating a hardening (see Appendix A) (Figure 1C). These observations were supported using the WLI, as measurements of the volume of individual droplets over time revealed three different evaporation rates (Figure 10).

Although no chemical analyses of the tarsal fluid of *M. extradentata* are readily available, of the composition of the fluids of other insect species allows us to draw assumptions on the composition based on the morphological observations (Figure 10).

Previous research on the adhesive fluid of insects with smooth pads (stick insects, cockroaches, and ants) also showed that it is a two-phase microemulsion, which consists of a volatile hydrophilic phase and a non-volatile hydrophobic phase [12,71,72].

The fast evaporation rate could be a result of the fraction of the tarsal fluid that has a potentially high volatile content (e.g., short-chained hydrocarbons and alcohols). A slow evaporation rate can be indicative of a droplet type that contains a higher proportion of non-volatile components (e.g., long-chained hydrocarbons and fatty acids) and a lower proportion of volatile components. The non-evaporating part of the tarsal liquid might consist of hardening non-volatile components (Figure 10 and Figure 11) [27,28,34,39,40,42,47,73].

Similar measurements were conducted by Peisker and Gorb (2011), where they measured the evaporation rates of individual tarsal adhesive secretion droplets of the hairy attachment systems of the fly *Calliphora vicina* and the beetle *Coccinella septempuctata* with an atomic force microscope over a time span of 60 min [73]. Within this time span, they discovered fast evaporation for the tarsal secretion of the fly and a comparably slower evaporation for the tarsal secretion of the beetle. The main difference in our findings for the secretion on the smooth attachment devices of *M. extradentata* is the presence of a range of different evaporation rates including both fast and slow evaporation rates in different droplets from the same footprint. As Peisker and Gorb measured the evaporation in the first 60 min, and we observed the evaporation for up to 80 days with larger time spans between measurements, a more precise comparison within the same time scale would be necessary for these examples. However, the tarsal secretions from the hairy attachment systems investigated therein seem to consist of more uniform droplets compared to the smooth attachment system of *M. extradentata*.

Our microscopy observations and experiments reported above allow us to make predictions about the material properties and behavior of the four tarsal secretion components. We may hypothesize that one part of the tarsal secretion being fluid and solidifying over time in the light microscope is the same component that shows no evaporation and the flake-like appearance in the cryo-SEM. The morphological indications for this hypothesis are the particular layered surface structure of sputtered flakes resembling the structure of dried fluid (Figure 5B,E,F), and the position of the flakes on the edge of the imprints, since this is the site where the majority of the other contaminants was located (Figure 5A). Chemical analyses of the tarsal secretion of other insects also found lipid-soluble components that could solidify over time [28,34]. Also, flake-like structures were previously detected in the footprints of *Locusta migratoria* [28]. Therefore, we may assume that the flakes are hardened and accumulate old parts of tarsal secretion, which are removed via the passive self-cleaning mechanism to the margin of the pad. The occurrence of flakes indicates that the adhesive fluid is at least partially composed of non-volatile components.

### 4.2. Self-Cleaning Mechanism

The passive self-cleaning mechanism of smooth attachment devices is important for attachment maintenance, as it removes contaminants that would reduce adhesive performance [30,31]. Contaminants reduce attachment by increasing the distance between the adhesive pad and the substrate, which, in turn, reduces the actual surface area that is available for contact with the substrate [31]. Two different kinds of contaminants were observed in the footprints: the old, hardened tarsal secretion in the form of flakes (Figure 5) and the contamination from the environment (Figure 9).

Clemente and colleagues described the effect of the tarsal secretion for self-cleaning in the stick insect *Carausius morosus*. They discovered that a high amount of tarsal fluid increases the recovery rate of the adhesion and hypothesized that this is due to the liquid filling the gaps [30,31]. We can confirm with our cryo-SEM data that the tarsal fluid supports the self-cleaning mechanism.

We observed that all contaminants were either covered or surrounded by the tarsal secretion (Figure 9A (surrounded) and B–D (covered)). Small particles are agglomerated together via the adhesive secretion [74], reducing the ratio of volume to surface area (Figure 9B–D), thereby enabling an agglomerate to be removed easier. Due to the convex shape of the adhesive pads, as well as the pressure of the newly produced tarsal secretion, both types of contaminants are transported further to the edge in subsequent steps during locomotion and are finally deposited on the substrate via the shearing motions of the tarsus (Figure 3A,B and Figure 5A).

### 4.3. Attachment

Various experiments showed that the tarsal secretion has different effects on the attachment force generation, ranging from implementing viscous and capillary forces [4,33] to leveling the asperities on the substrate surfaces [48,50]. Our observations on the morphology of the secretion components provide support for the understanding of the following effects of the secretion in *M. extradentata*.

### 4.4. Viscosity and Capillary Forces

Multiple experiments indicate that the action of the capillary and viscous forces is important to generate adhesion in wet contacts [32,33,35,54,75]. We observed different morphological manifestations of the same fluid, which must vary in viscosity, and thus may affect the viscous and capillary forces to different extents.

The thick film components display morphological characteristics that can be attributed to those of viscous fluids, such as covering a large surface area (Figure 5A,E) and possessing a large volume (Figure 5 (dark grey scale)). We also measured droplets exhibiting a slow evaporation rate and appearing to be more viscous than others (Appendix A). Besides viscosity, the surface tension influences the liquids’ interaction with the substrate and can contribute to them having similar appearances. However, viscosity and surface tension are somewhat related [76,77].

Therefore, the thick films are likely rather viscous and/or possess a comparably high surface tension, which potentially aids in implementing the viscous force at the tarsus–substrate interface.

The droplet components show different morphology than the films, as they form individual roughly round shapes that can accumulate into big complexes. We found droplets in the footprints exhibiting a fast evaporation rate (mean evaporation rate of 8.629 ± 7.503%/day and 0.00599 ± 0.00521%/min) (Figure 10C). Due to the spectrum of different morphologies and evaporation rates occurring in the same secretion, the adhesive fluid can likely adapt to different substrate qualities and thus effectively combine capillary and viscous forces, enhancing attachment performance.

### 4.5. Free Surface Energy

For insects, the adaptation to different free surface energies of the substrates is challenging, as this is a factor that can influence locomotion, which, for example, plants utilize to either repel or capture insects [78,79,80]. A chemical analysis of the tarsal fluids of insects has shown that they are composed of hydrophilic and hydrophobic components in an emulsion that is capable of adapting to different free surface energies [28,34]. Although our morphological observations are not sufficient to predict the chemical composition of the footprints, the different morphological appearances of the fluids in the same footprint allow us to speculate about the physical properties of the components.

Nano-droplets were found along with the larger droplets (Figure 4D and Figure 8E), supporting the previous findings of a highly viscous emulsion consisting of lipid droplets in a water-based solution [28]. Different ice crystal shapes were observed on the surfaces of the thick film components (Figure 8A,B,D), which were likely formed due to the freezing of the water that was present within these components.

The droplet and thick film components show a range of surface structures, which can be smooth, rough, or granular (Figure 4C,F (droplets) and Figure 8C,F (thick film)). This suggests that the chemical composition of these components could differ and that, overall, the different components consist of a mixture of the same compounds, which assemble into the morphology we observed. Further evidence for this hypothesis could be found in the specificity of evaporation rates (Figure 11A). Assuming that the different components consist of a mixture of the same ingredients with varying proportions within the composition, it appears likely that the broad spectrum of evaporation rates found in the droplets is a result of the volatile components’ decreasing proportions in droplets with slower evaporation rates. This would also explain why the evaporation rates of the non-evaporating and slowly evaporating droplets did not differ statistically (Figure 10B) in spite of different evaporation behaviors and visual differences under light microscopes (our own observations). Further evidence of the ability of the tarsal secretion to support the attachment on surfaces with different surface energies was shown in *M. extradentata*. In these experiments, *M. extradentata* was able to adhere to a highly hydrophobic surface (Polytetrafluoroethylene (PTFE)) even underwater [55].

Since different compositions of the adhesive fluid should respond differently to the free surface energy of the substrate, detailed analyses of the chemical composition are required for a deeper understanding of the functional role of the fluid in the tarsal attachment system of the stick insect.

### 4.6. Leveling Substrate Asperities

An insect’s adhesive secretion is often mentioned as having a function in leveling the surface roughness in the adhesive contact [32,51,81,82]. Rough surfaces cause a reduced contact area between the adhesive pad and the substrate. Tarsal adhesive secretions fill up gaps in the roughness, increasing the contact area and thus the attachment forces [48,49,53]. To fulfil this task, such secretion needs to cover large areas to be available in sufficient amounts, and their viscosity must be adapted to the corresponding roughness hierarchy. The presumable fluid components observed in the cryo-SEM showed different dewetting morphologies. Thin film components cover a wide gap-less area with a thin film (thin films cover 1 mm in diameter (Figure 6A)). Thick film components cover the surface with a patchy thick film (thick films cover 2 mm in diameter (Figure 7A,E)), and the droplet components form single droplets, which can accumulate into larger complexes (droplets cover ~200 µm in diameter (Figure 4)).

As the surfaces of natural substrates have fractal roughness at different hierarchical levels [83,84,85], a mixture of fluids with a range in viscosity would be helpful to quickly adapt to a range of roughness at once. Thick films are probably more viscous and, hence, are able to fill large gaps of coarse roughness. Thin films potentially have a lower viscosity, which is judged on their low volume and wide spreading on the substrates, and likely more readily fill gaps of finer roughness. The low volume is visible in the lower height of the latter fluids in imprints, which disappear completely when covered by a 10 nm Au-Pa layer when sputter-coated (see Appendix A). Another indication that the tarsal fluid of *M. extradentata* can have an influence on the leveling of the substrate asperities is the performance of *M. extradentata* on substrates with varying roughness, which was reported in previous experiments [62,68].

## 5. Conclusions

The cryo-SEM enabled an examination of the pad fluid in its frozen state immediately after deposition. We identified four morphologically different components that originate from the same tarsal secretion. The measurements of the evaporation rate of individual droplets indicate that the liquid consists of a spectrum of slowly evaporating to fast-evaporating components. These observations suggest that the tarsal fluid is a mixture of volatile and non-volatile components that, working in concert, extend the properties of the adhesive secretion.

Parts of the adhesive secretion can harden over time into flakes and thus contaminate the adhesive area of the attachment pads. Contaminations can be glued together via new adhesive fluid and can be passively removed via tarsal movement during locomotion. Due to the presence of morphologically and physically different components, the adhesive fluid can support different phases of attachment, including contact generation, contact maintenance, and contact breakage, for example, by filling varying degrees of roughness and generating capillary and/or viscous forces.

With this study, we show how the morphologically and physically diverse tarsal secretion of *Medauroidea extradentata* could potentially contribute to the range of functions. These results allow for several possible ideas to be generated for further investigations. A detailed chemical analysis of the adhesive secretion would aid in making a correlation between the morphological features and chemical composition. Histological studies could provide insights into the structure and distribution of the exocrine cells that are involved in the production of the secretion. Studies of the composition of the tarsal fluid of different ecologically specialized taxa can aid in understanding the adaptability of tarsal secretions. These insights could be valuable for the development of novel biomimetic adhesive fluids.

## Figures and Tables

**Figure 1 biomimetics-08-00439-f001:**
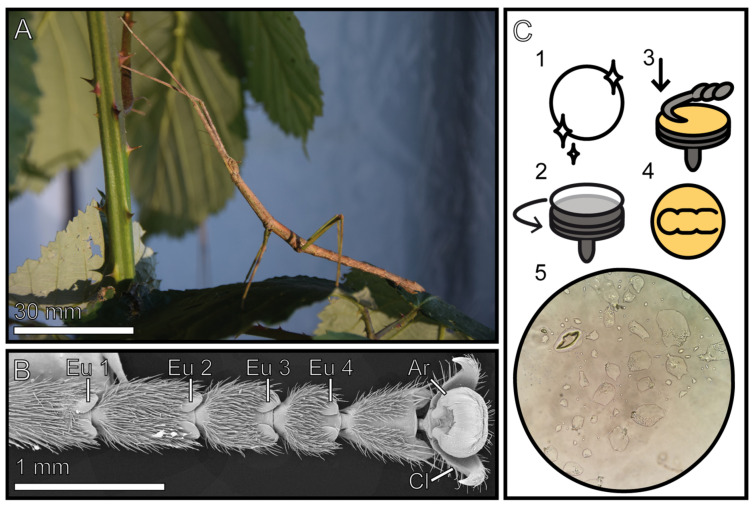
Experimental setup. (**A**) Adult female *Medauroidea extradentata*. (**B**) Overview of the tarsus of *M. extradentata*; Ar = arolium, Cl = claw, Eu = euplantulae. (**C**) The sequence of footprint generation: round glass coverslip is cleaned (1), glass cover slip is mounted on a cryo-SEM stub (2), cover slip is sputtered, tarsus of living insect is positioned on top, and pressure is applied (arrow) (3), footprint remains on the glass (4), footprint under a light microscope (5). (**C**) Reproduced with permission from Thomas et al. (2023) [55]. Copyright: The Company of Biologists.

**Figure 2 biomimetics-08-00439-f002:**
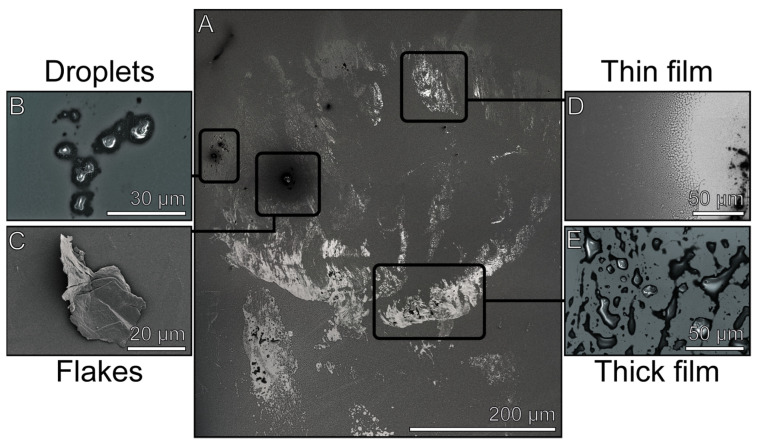
Footprint overview containing the four main components. (**A**) Arolium footprint of *M. extradentata*. Examples of droplets (**B**), flakes (**C**), thin films (**D**), and thick films (**E**).

**Figure 3 biomimetics-08-00439-f003:**
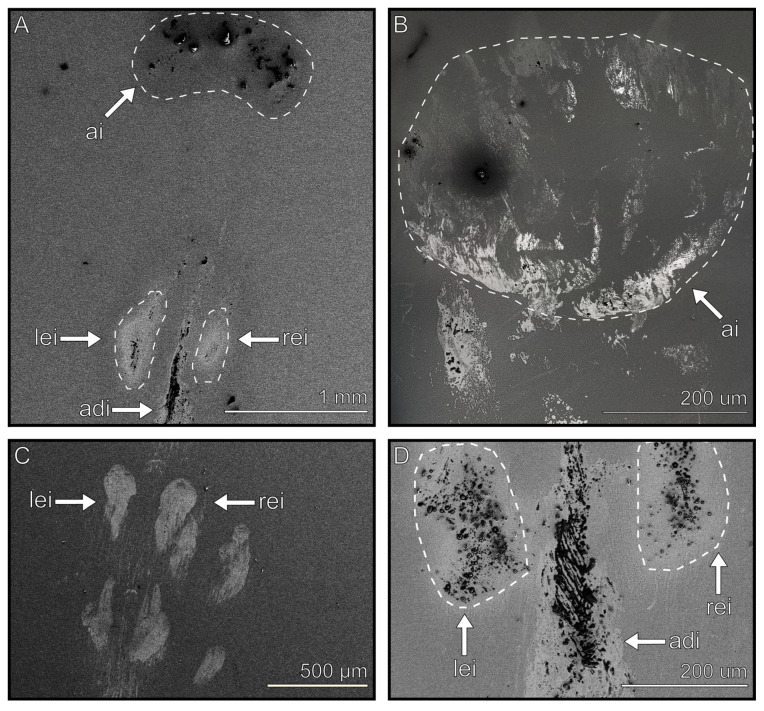
*M. extradentata* footprints from different pads. (**A**) Arolium imprint visible in the top half and imprint of first euplantulae in the lower half, with an additional imprint between them. (**B**) Arolium imprint. (**C**) Two euplantulae imprints below one another. (**D**) Euplantulae imprints with an additional tarsal secretion imprint between them. Outlines highlight the edges of the imprint. ai = arolium imprint, lei = left euplantulae imprint, rei = right euplantulae imprint, adi = additional imprint.

**Figure 4 biomimetics-08-00439-f004:**
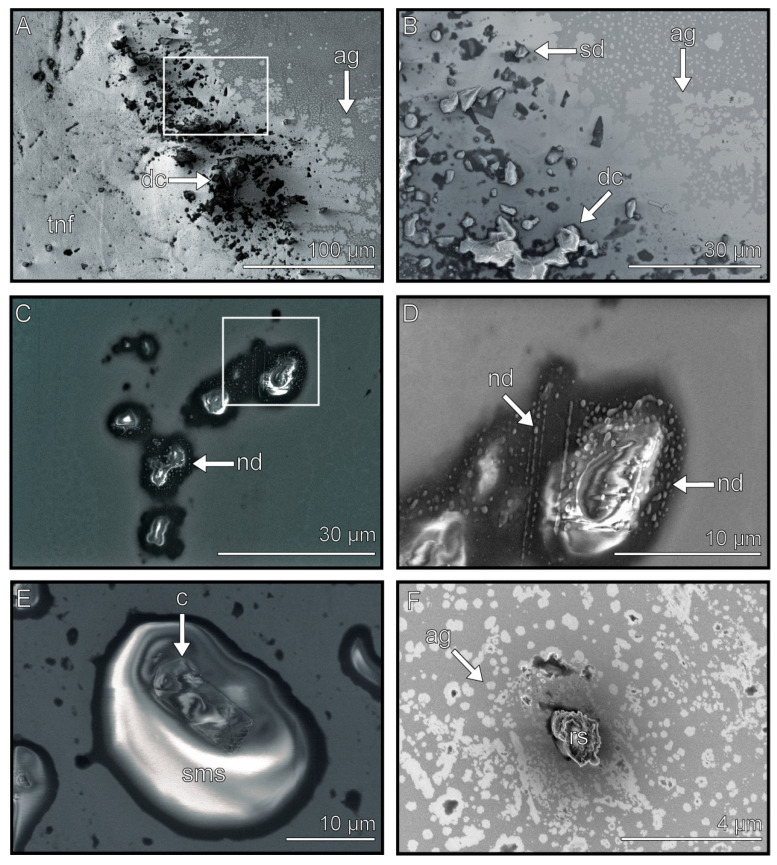
Droplets in the secretion. (**A**) Overview of an imprint, which mainly consists of droplets and thin films. The box displays the location that image (**B**) originates from. (**B**) Droplets can occur as single droplets, or they can accumulate into complexes. (**C**) Droplets with nano-droplets on and around them. The box displays the location where image (**D**) originates from. (**D**) Single droplet with magnified view on the nano-droplets. (**E**) Single droplet with a smooth surface structure and an absorbed contamination particle. (**F**) Single droplet with a rough surface and aggregated thin film around it. sd = single droplet, dc = droplet complex, ag = thin film aggregate, nd = nano-droplets, c = contamination, sms = smooth surface, rs = rough surface, tnf = thin film.

**Figure 5 biomimetics-08-00439-f005:**
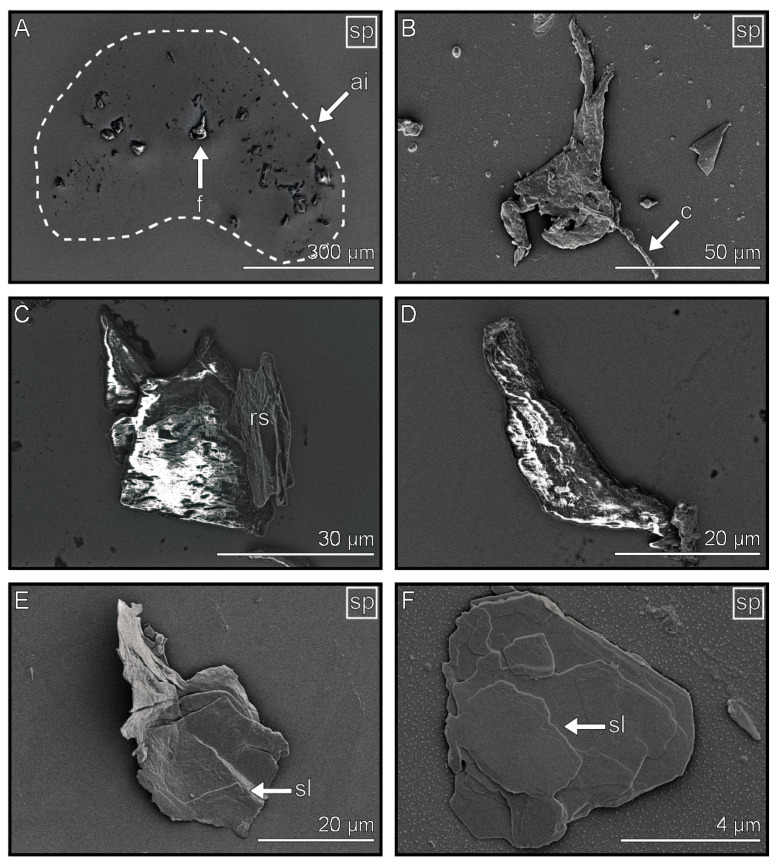
Flakes in the tarsal secretion. (**A**) Distribution pattern of the flakes in an arolium imprint. (**B**–**F**) Exemplary shapes, sizes, and surface structures of the flakes. (**B**) Elongated sputtered flake with adsorbed contamination. (**C**) Compressed unsputtered flake. (**D**) Elongated unsputtered flake. (**E**) Sputtered flake with layered surface structure. (**F**) Small and sputtered flake with layered surface structure. Outlines highlight the edges of the imprint. Images marked with “sp” show sputtered samples. ai = arolium imprint, f = flake, c = contamination, sl = surface layer, rs = rough surface structure.

**Figure 6 biomimetics-08-00439-f006:**
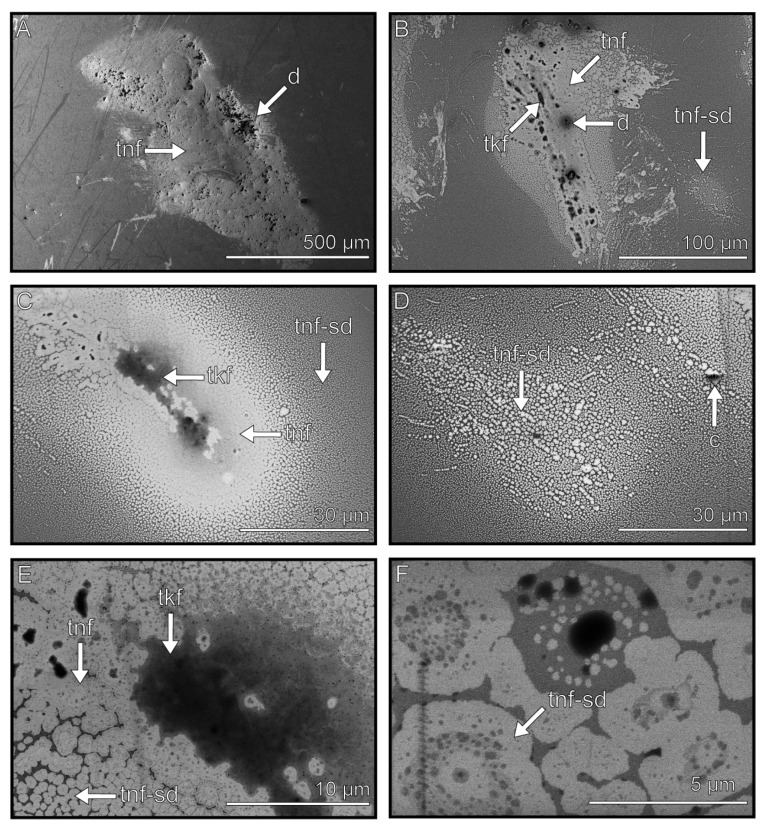
Thin films in the tarsal secretion. (**A**) Overview of a large imprint with a high proportion of the thin film. (**B**) Imprint with thin film and thick film components. The thin film forms small aggregates in the periphery of the other components. (**C**) When the thin film is closer to other components, it forms a uniform surface, whereas when it is distant to other components, it forms progressively smaller aggregates. (**D**) In absence of other components, small round aggregates are formed. (**E**,**F**) Higher magnification of a uniform thin film (**E**) and a small aggregate (**F**). tnf = thin film, d = droplet, tkf = thick film, tnf-sd = thin film small droplet, c = contamination.

**Figure 7 biomimetics-08-00439-f007:**
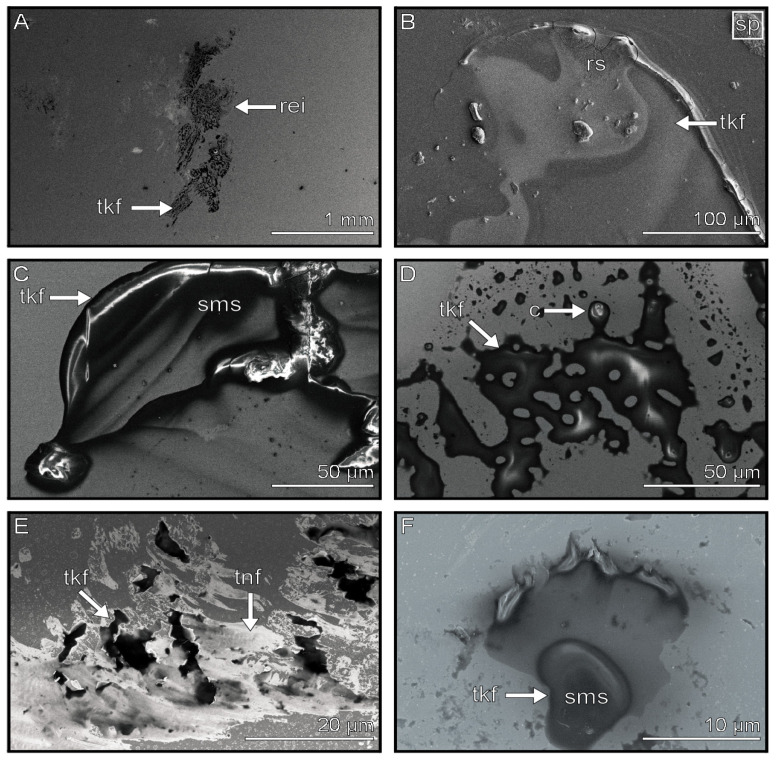
Thick films in the tarsal secretion. (**A**) Imprint mainly consists of thick film components. (**B**) Uniform sputtered thick films with rough surfaces. (**C**) Uniform imprint with a smooth surface structure. (**D**) Thick film imprint with gaps and a smooth surface. (**E**) Transition between uniform thick film and thin film residuals. (**F**) Thick film with smooth surface structure. Images marked with “sp” show sputtered samples. rei = right euplantulae imprint, tkf = thick film, sms = smooth surface structure, c = contamination, tnf = thin film, rs = rough surface structure.

**Figure 8 biomimetics-08-00439-f008:**
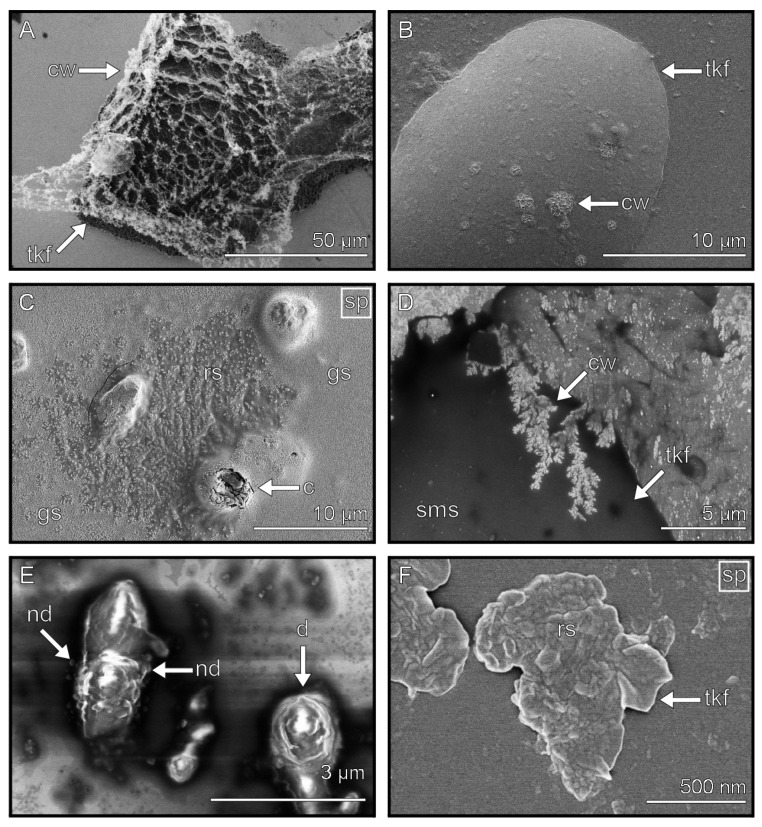
Different surface structures of tarsal secretion components. (**A**) Thick film component covered by a mesh of frozen water. (**B**) Thick film component with multiple single patches consisting of ice crystals. (**C**) Close-up of a sputtered thick film with rough and granular surface and enclosed contamination. (**D**) Thick film with ice formation on its smooth surface. (**E**) Droplet component with nano-droplets. (**F**) Rough surface of sputtered thick film. Images marked with “sp” show sputtered samples. cw = crystallized water (ice), tkf = thick film, rs = rough surface structure, gs = granular surface structure, smooth surface structure, c = contamination, nd = nano droplets, d = droplet.

**Figure 9 biomimetics-08-00439-f009:**
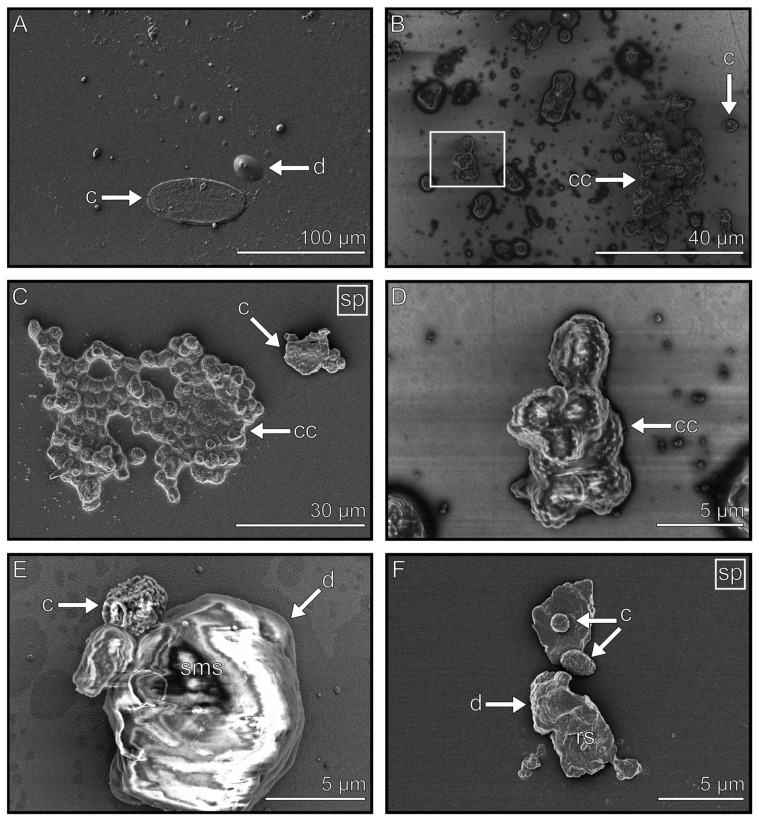
Fluid interactions with contaminants. (**A**) Large contamination covered and surrounded by tarsal secretion. (**B**,**C**) Multiple contaminants clustered by the tarsal secretion (unsputtered (**B**) and sputtered (**C**)). Box in (**B**) displays the magnified region of (**D**). (**D**) Close-up of contaminants clustered by tarsal secretion. Single contamination adhered to the surface of a droplet (unsputtered (**E**) and sputtered (**F**)). Images marked with “sp” show sputtered samples. c = contamination, d = droplet, cc = contamination cluster, sms = smooth surface structure, rs = rough surface structure.

**Figure 10 biomimetics-08-00439-f010:**
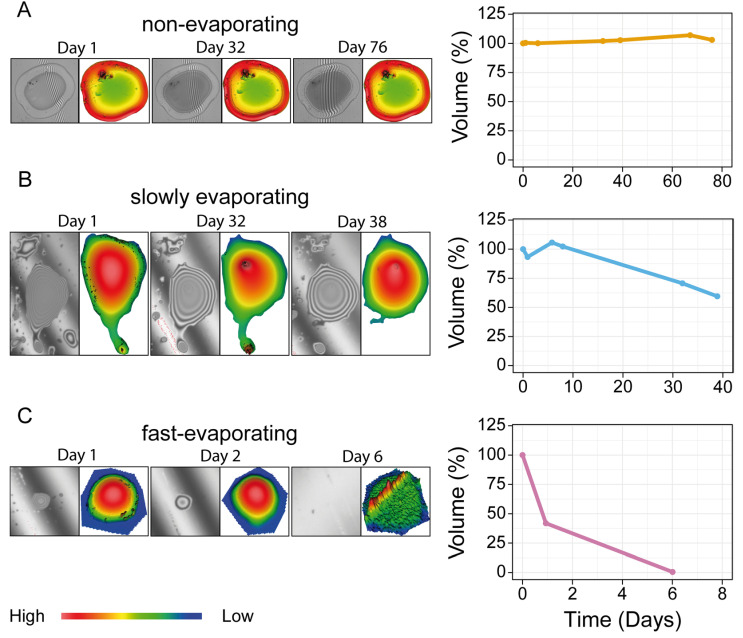
WLI measurements of droplet evaporations. Examples of non-evaporating (yellow) (**A**), slowly evaporating (blue) (**B**), and fast-evaporating (pink) droplets (**C**). On the left side, droplets are shown at three different measurement days for each droplet type. For every droplet, a microscopy image of the droplet taken with the WLI microscope (left) and the corresponding 3D heatmap showing its volume (right) are given. Graphs show measurement curves of the corresponding curves. The colors of the 3D images represent the relative height of the droplets: red = highest part; blue = lowest part.

**Figure 11 biomimetics-08-00439-f011:**
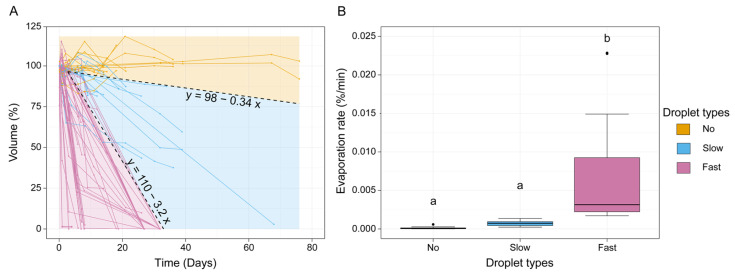
Change in droplet volume (in %) during 78 days of experiment (**A**) and evaporation rates (in %/min) (**B**). (**A**) Change in droplet volume. Droplet types are color-coded (non-evaporating droplets = yellow; slowly evaporating droplets = blue; fast-evaporating droplets = pink) and the regression lines of the boundaries to the three evaporation rate types are represented with corresponding linear regression equations. (**B**) Evaporation rates (in %/min) of non-evaporating droplets (yellow, *n* = 14), slowly evaporating droplets (blue, *n* = 12), and fast-evaporating (pink, *n* = 44) droplets. The values correspond to the mean evaporation rate (in %/min) of each individual droplet. Groups with different lowercase letters are statistically different (Kruskal–Wallis one-way ANOVA on ranks, *p* < 0.001 with Dunn’s post hoc test, *p* < 0.05). Boxes indicate the 25th and 75th percentiles, whiskers are the 10th and 90th percentiles, and the line within the boxes shows the median.

## Data Availability

The raw data of the evaporation measurements (Appendix A) and an additional SEM picture (Appendix A) and light microscopy videos (Appendix A) are available as Appendix A.

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
