# Peer review of "Characterization of Morphologically Distinct Components in the Tarsal Secretion of Medauroidea extradentata (Phasmatodea) Using Cryo-Scanning Electron Microscopy"

_biomimetics, 2023, doi:10.3390/biomimetics8050439_

Round 1

Reviewer 1 Report

The manuscript presents an interesting morphological study on the tarsal secretion of a stick insect, the Medauroidea extradentata (Phasmatodea), using cryo-scanning electron microscopy and white light interferometry. white light interferometry.

This study intends to take a step forward in the understanding and potential development of multifunctional biomimetic adhesive structures, featuring adaptive features.

The manuscript is overall well-written.

The introduction and materials and methods parts are clear.

The results are correctly described in the text, and the various supporting figures are of good quality.

The discussion section is generally good. However, it does not take full advantage of the authors' earlier complementary results on the same insect species, published and cited here, nor on other existing literature [complementary] data.

The conclusion reflects the rather good but incomplete discussion of the experimental results,

in the light of already published complementary work and other complementary literature data.

The different questions still to be answered are very pertinent.

In the meantime, some paragraphs should be reformulated, such as:

1° Lines 522-525 and 582-585: The paragraphs below should be reformulated.

 ["The thick film components display morphological characteristics of viscous fluids, such as covering a large surface area (Figure 5 A and E) and possessing a large volume (Figure 5 (dark grey scale)), that must correspond to their different physical characteristics."]

and

["Therefore, the thick films are likely rather viscous and probably aid in implementing viscous force at the tarsus-substrate interface."]

In fact, for a given surface (of a specific chemical nature), the contact area also depends on the surface tension, and not just on the fluid's (potentially high) viscosity. It is therefore not entirely appropriate to consider large surface area and volume as intrinsic characteristics of a viscous fluid.

Surface tension will also depend on the chemical composition of the fluid, the affinity of certain parts of the fluid with the surface of the "substrate", the mobility of these parts within the fluid, and so on.

2° Lines 453-457: The factual observation in this paragraph should be followed by a pertinent discussion or interpretation.

["The fast evaporation rate corresponds to the portion of the tarsal liquid with a high proportion of volatile components. Slow evaporation rate indicates the portion of the tarsal liquid with a high amount of non-volatile and probably lipid-like components. Non-evaporating part of the tarsal liquid is 456 composed of hardening non-volatile components (Figure 10; 11).”]

Otherwise, it is rather predictable that volatile components will evaporate first and quickly, so fluids with higher amounts of volatile components will see their volume and contact size decreasing faster, compared with fluids with less or no volatile components. Besides, different volatile compounds have different evaporation rates, depending on their chemical nature, molecular size… The evaporation rate could be influenced somehow by the mobility of these volatile components within the fluid volume (depending on the fluid viscosity and film thickness, etc.).

In addition, more explicit information about the chemical nature of the typical volatile components in such films would be necessary [at least general information from literature data].

3° Minor observations:

- Lines 167, 458: the years of publication should be replaced by the reference numbers.

One other aspect which is not fully clear and explained concerns the very large number of self-citations (31 from a total of 70 references), several on the same species of insect, that it becomes difficult to properly assess the real step forward and original contribution of the results obtained in this work.

For all these elements, I recommend a major revision of the manuscript.

Author Response

For research article

Response to Reviewer 1 comments

Dear Reviewer 1,

Thank you very much for taking the time to review our manuscript. We have looked at your suggestions for improvement and made the following changes.

In the following we list the answers to all comments by you and how we followed them. We followed all the suggestions. The corresponding corrections are highlighted in tracked changes in the re-submitted file.

Sincerely, on behalf of all authors,

Julian Thomas

Department of Functional Morphology and Biomechanics, Zoological Institute,

Kiel University, 24118 Kiel, Germany

Comment:

However, it does not take full advantage of the authors' earlier complementary results on the same insect species, published and cited here, nor on other existing literature [complementary] data.

Answer:

We have added a deeper insight into the adhesive secretion systems of other animal groups:

Lines 45-57: Except for insects, other animal groups also produce a fluid onto the substrate to support the adhesion process [14–16]. However, the mechanisms can largely differ from those employed by insects: Tree frogs similarly, but convergently, produce an adhesive liquid in their mucus glands to strengthen their attachment [17,18]. Geckos do not produce large amounts of tarsal secretion, but it has been shown that a nanometer thin lipid film covers the surface of their adhesive setae [19]. Especially in marine environments other depletion mechanisms are found: Echinoderms for example secrete a two-phasic secretion, where the first phase generates the adhesion and the second phase dissolves the first phase [20–23]. Freshwater polyps (Hydra spp.) can achieve temporary adhesion based on their adhesive secretion which consists mainly of fibres [24]. The fluid depletion of stick and leaf insects in contrast works differently: it is most likely a passive delivery mechanism of tarsal fluid from the adhesive pads onto the substrate facilitated by pressure and intermolecular forces [12]

Discussed the results of this work with the results of our previous work on M. extradentata

Lines 636-639: Further evidence of the ability of the tarsal secretion to support the attachment on surfaces with different surface energies was shown in M. extradentata. In these experiments M. extradentata was able to adhere to a highly hydrophobic surface (Polytetrafluoroethylene (PTFE)) even under water [55].

Lines 665-669: Another indication that the tarsal fluid of M. extradentata can have an influence on levelling the substrate asperities is the performance of M. extradentata on substrates with varying roughness reported in previous experiments [62,68].

Comment:

1° Lines 522-525 and 582-585: The paragraphs below should be reformulated.

 ["The thick film components display morphological characteristics of viscous fluids, such as covering a large surface area (Figure 5 A and E) and possessing a large volume (Figure 5 (dark grey scale)), that must correspond to their different physical characteristics."]

and

["Therefore, the thick films are likely rather viscous and probably aid in implementing viscous force at the tarsus-substrate interface."]

In fact, for a given surface (of a specific chemical nature), the contact area also depends on the surface tension, and not just on the fluid's (potentially high) viscosity. It is therefore not entirely appropriate to consider large surface area and volume as intrinsic characteristics of a viscous fluid.

Surface tension will also depend on the chemical composition of the fluid, the affinity of certain parts of the fluid with the surface of the "substrate", the mobility of these parts within the fluid, and so on.

Answer:

Discussed in more detail the possible influence of surface tension on morphological characteristics and linked it to viscosity.

Lines 591-601: The thick film components display morphological characteristics that can be attributed to those of viscous fluids, such as covering a large surface area (Figure 5 A and E) and possessing a large volume (Figure 5 (dark grey scale)) We also measured droplets exhibiting a slow evaporation rate and appearing more viscous than others (supplementary video S2). Besides viscosity, surface tension influences the liquids’ interaction with the substrate and can contribute to similar appearance. However, viscosity and surface tension are somewhat related [76,77]. Therefore,the thick films are likely rather viscous and/or possess a comparably high surface tension, which potentially aids in implementing viscous force at the tarsus-substrate interface.

Comment:

2° Lines 453-457: The factual observation in this paragraph should be followed by a pertinent discussion or interpretation.

["The fast evaporation rate corresponds to the portion of the tarsal liquid with a high proportion of volatile components. Slow evaporation rate indicates the portion of the tarsal liquid with a high amount of non-volatile and probably lipid-like components. Non-evaporating part of the tarsal liquid is 456 composed of hardening non-volatile components (Figure 10; 11).”]

Otherwise, it is rather predictable that volatile components will evaporate first and quickly, so fluids with higher amounts of volatile components will see their volume and contact size decreasing faster, compared with fluids with less or no volatile components. Besides, different volatile compounds have different evaporation rates, depending on their chemical nature, molecular size… The evaporation rate could be influenced somehow by the mobility of these volatile components within the fluid volume (depending on the fluid viscosity and film thickness, etc.).

In addition, more explicit information about the chemical nature of the typical volatile components in such films would be necessary [at least general information from literature data].

Answer:

Discussed in more detail the possible causes of the different evaporation rates and linked them to the chemical analyses of other adhesion secretions.

Lines 515-526: Although no chemical analyses of the tarsal fluid of M. extradentata are readily avaiable, of the composition of the fluids of other insect species allows to draw assumptions on the composition based on the morphological obsevations (Figure 10). Previous research on the adhesive fluid of insects with smooth pads (stick insects, cockroaches and ants) also showed that it is a two-phasic microemulsion which consists of a volatile hydrophilic and a non-volatile hydrophobic phase [12,71,72]. The fast evaporation rate could be a result of the fraction of the tarsal fluid that has a potentially high volatile content (e.g. short-chained hydrocarbons and alcohols). A slow evaporation rate, can be indicative for a droplet type that contains a higher proportion of non-volatile components (e.g. long-chained hydrocarbons and fatty acids) and a lower proportion of volatile components. The non-evaporating part of the tarsal liquid might consist of hardening non-volatile components (Figure 10; 11) [27,28,34,39,40,42,47,73].

Comment:

Lines 167, 458: the years of publication should be replaced by the reference numbers.

Answer:

Was adapted to the citation style of the manuscript

Comment:

One other aspect which is not fully clear and explained concerns the very large number of self-citations (31 from a total of 70 references), several on the same species of insect, that it becomes difficult to properly assess the real step forward and original contribution of the results obtained in this work.

Answer:

By adding the additional information and expanding the discussion, the proportion of own citations has been reduced.

Reviewer 2 Report

This study is interesting and clearly presented. The authors do a good job demonstrating the complexity of these secretions, and this study will inform future work in the area. I only have minor suggestions.

The authors sometimes use the term “fluid depletion” (lines 100, 199) when referring to the deposition of the droplet. This expression should be defined clearly, since it is different from how adhesive deposition is often described in other animals.

I didn’t see a description of the conditions used when the authors measured the evaporation rate. What was the temperature and relative humidity? Was it in a closed chamber?

Line 207-8: When referring to the gray scale for the cryo-SEM images, the authors explain why some regions are darker, but there are also areas that are actually brighter than the background. What does that mean?

Line 270: “…appear to consist of dried liquid” – on what basis is this conclusion made?

Lines 449-451: qualitative observations of viscosity should be in results.

Author Response

For research article

Response to Reviewer 2 comments

Dear Reviewer 2,

Thank you very much for taking the time to review our manuscript. We have looked at your suggestions for improvement and made the following changes.

In the following we list the answers to all comments by you and how we followed them. We followed all the suggestions. The corresponding corrections are highlighted in tracked changes in the re-submitted file.

Sincerely, on behalf of all authors,

Julian Thomas

Department of Functional Morphology and Biomechanics, Zoological Institute,

Kiel University, 24118 Kiel, Germany

Comment:

The authors sometimes use the term “fluid depletion” (lines 100, 199) when referring to the deposition of the droplet. This expression should be defined clearly, since it is different from how adhesive deposition is often described in other animals.

Answer:

The term "fluid depletion" was described in more detail and placed in the context of adhesive secretion systems of other animal groups.

Lines 45-57: Except for insects, other animal groups also produce a fluid onto the substrate to support the adhesion process [14–16]. However, the mechanisms can largely differ from those employed by insects: Tree frogs similarly, but convergently, produce an adhesive liquid in their mucus glands to strengthen their attachment [17,18]. Geckos do not produce large amounts of tarsal secretion, but it has been shown that a nanometer thin lipid film covers the surface of their adhesive setae [19]. Especially in marine environments other depletion mechanisms are found: Echinoderms for example secrete a two-phasic secretion, where the first phase generates the adhesion and the second phase dissolves the first phase [20–23]. Freshwater polyps (Hydra spp.) can achieve temporary adhesion based on their adhesive secretion which consists mainly of fibres [24]. The fluid depletion of stick and leaf insects in contrast works differently: it is most likely a passive delivery mechanism of tarsal fluid from the adhesive pads onto the substrate facilitated by pressure and intermolecular forces [12]

Comment:

I didn’t see a description of the conditions used when the authors measured the evaporation rate. What was the temperature and relative humidity? Was it in a closed chamber?

Answer:

The exact environmental conditions for the evaporation measurements were added.

Lines 170-172: he glass slides were stored in a closed glass chamber at 20.6 – 22.9 °C room temperature and 43.2 – 51.4 % ambient humidity (measured within the chamber).

Lines 191-192: The glass slides were placed under the WLI and measured at 20.6 – 22.9 °C room temperature and 61 – 64.3 % ambient humidity.

Lines 212-217: 2.5 Temperature and ambient humidity measurements:

Temperature and the ambient humidity were measured with a Tinytag Plus 2 TGP – 4500 (Gemini Data Loggers, Chichester, UK) and analyzed using the software TinyTag Explorer 6.0 (Gemini Data Loggers, Chichester, UK). For the measurements in the closed glass chamber, 250 measurements at an interval of 30 min and for the measurements at the WLI 200 measurements at an interval of 1 min were conducted.

Comment:

Lines 207-8: When referring to the gray scale for the cryo-SEM images, the authors explain why some regions are darker, but there are also areas that are actually brighter than the background. What does that mean?

Answer:

A more detailed description of the gray scale of the cryo-SEM images was added.

Lines 232-234: Accordingly, thin liquid layers with a low electron density are displayed brighter than the background, as well as the thick layers with a high electron density that appear dark (Figure 2).

Comment:

Line 270: “…appear to consist of dried liquid” – on what basis is this conclusion made?

Answer:

The part was removed as it is explained in more detail in the discussion.

Lines 313-315: After sputtering, the flakes show a structured surface consisting of several parallel thin layers with a thickness of a few nanometers, which do not have a uniform shape

Lines 504-505: The varying viscosities were detected by drawing a fine needle through individual droplets (see Light microscopy observations).

Comment:

Lines 449-451: qualitative observations of viscosity should be in results

Answer:

The observations of the droplets under the light microscope were added to the Results.

Lines 445-453: 3.9 Light microscopy observations

Randomly selected glass slides with deposited footprints were observed and filmed at different time intervals under an inverted microscope. During the observation, a fine needle was carefully pulled through individual droplets and the different behavior of the droplets was observed and filmed (supplementary videos S1 and S2). Two different droplet responses were observed: 1) droplets that appeared to be liquid and were split into smaller droplets by the needle (supplementary video S1), and 2) droplets that appeared to be more viscous, whereby the needle scratched their surface (supplementary video S2).

Round 2

Reviewer 1 Report

The revised manuscript is much improved.

I recommend it for publication in the present form..